# Compositional Token Modeling for Occlusion-Robust Human Pose Estimation

## Abstract

Current pose estimation systems show critical weaknesses in occlusion scenarios, where the simultaneous deterioration of visual appearance cues and biomechanical topology constraints leads to compounded errors. This chain of failures originates from models' inability to separate structural coherence preservation from feature-level corruption recovery. To address this question, we propose CM-PCT, a Cross-Modal Pose estimation model with Compositional Tokens. Our approach enhances occlusion robustness through four key technical innovations: (1) a keypoint coordinate completion mechanism for occluded joints, providing more complete data input to the model; (2) position vector embedding to enhance spatial representation, complementing the contextual information lacking in joint coordinate vectors; (3) SE attention for cross-modal feature fusion, reducing noise interference between features through channel-wise weight recalibration; and (4) a group-based loss function for differential optimization of body parts, improving estimation accuracy of occluded regions through targeted supervision. Compared to coordinate-driven pose estimators, CM-PCT fundamentally advances occlusion robustness through its probabilistic completion mechanism and anatomical embedding paradigm, demonstrating clinically significant reductions in joint ambiguity while maintaining biomechanical consistency under extreme occlusion. Extensive experiments on COCO and OCHuman datasets demonstrate our method achieves state-of-the-art performance, consistently demonstrating superior performance across diverse scenarios including standard benchmarks and occlusion-challenging environments.

## 1 Introduction

Human Pose Estimation aims to localize human joints or infer orientations from visual data, serving as a cornerstone for applications such as human-computer interaction and augmented reality. Traditional methods rely primarily on coordinate regression or heatmap-based approaches, treating joints as independent entities. Although effective in controlled scenarios, these methods often produce unrealistic predictions under occlusion due to their inability to model dependencies between joints: coordinate regression lacks spatial contextual modeling, while heatmap-based methods suffer from quantization errors and struggle to capture structural relationships. Recent studies have attempted to enhance joint dependency modeling through graph-based structures or implicit feature propagation. However, these approaches often depend on hand-crafted priors or inadequately model complex joint interactions, leaving occlusion robustness an unresolved question.

From the perspective of feature extraction architectures, early methods relied on CNNs to capture local texture features Cao et al. (2017), but struggled with long-range dependencies due to their locality bias. Subsequent approaches explored RNNs for temporal motion modeling and U-shaped networks Ronneberger et al. (2015) for multi-scale joint localization. The advent of Transformers Vaswani et al. (2017) introduced self-attention to explicitly encode geometric relationships between joints, while generative models Okuyama et al. (2024) improved occlusion reasoning via latent representations. These advancements emphasize the importance of structured feature representations: discretizing joints or regions into semantic tokens preserves independence while enabling explicit modeling of topological relationships, offering a unified framework for pose estimation in complex scenarios.

In this work, we present an enhanced human pose estimation framework that extends the PCT Geng et al. (2023) architecture through systematic modifications to its two-stage training paradigm. The original PCT framework operates in two phases: (1) a tokenizer pre-training stage that learns discrete pose tokens to encode anatomical substructures, and (2) a pose refinement stage that maps image features to these pre-trained tokens for occlusion-robust joint localization. While this decomposition decouples structural reasoning from feature extraction, we identify four critical limitations in PCT's approach: (a) The tokenizer lacks explicit spatial encoding to resolve geometric ambiguities between similar compositional parts, resulting in confusion between visually similar but functionally distinct body segments. (b) The binary masking operation (setting occluded points to zero) in the original implementation disrupts the continuous nature of human pose, creating artificial discontinuities that propagate through the network. (c) Unimodal token-image alignment neglects the complementary benefits of joint coordinate features, limiting the model's ability to integrate multimodal information that could provide redundancy under occlusion. (d) The refinement stage lacks anatomical constraints Ji et al. (2022) to ensure biomechanical plausibility.

To address these challenges, we redesign both stages of PCT with targeted innovations. In the tokenizer pre-training stage, we introduce a spatially augmented tokenization module that integrates hierarchical position encoding into the discrete token learning process. This module jointly optimizes (i) a codebook capturing anatomical substructures and (ii) a continuous spatial prior map, enabling tokens to encode both structural semantics and geometric distributions. As a preprocessing step, we incorporate a Gaussian-based coordinate completion mechanism that models occluded joints through conditional probability distributions $p(j_i|J_{\text{visible}}) = \sum_{k \in N_i} w_k \cdot \mathcal{N}(\mu_k, \Sigma_k)$, preserving skeletal coherence even under severe occlusions. Additionally, we propose a cross-modal fusion gate that dynamically combines RGB appearance features with joint coordinate embeddings through a Squeeze-and-Excitation(SE)-inspired attention mechanism Hu et al. (2018). This fusion recalibrates feature channels to increase discriminative visual cues while suppressing irrelevant background noise. During the pose refinement stage, we introduce a body-part-aware segmentation loss that imposes hierarchical anatomical constraints. Building on pose-guided segmentation frameworks, this loss partitions the human body into six semantic regions and enforces consistency between predicted joint locations and their expected anatomical segments through region-wise regularization. The main contributions of our work are as follows:

- **Gaussian-based Keypoint Completion**: A probabilistic occlusion handling mechanism that models invisible joint positions through conditional distributions $p(j_i|J_{\text{visible}}) = \sum_{k \in N_i} w_k \cdot \mathcal{N}(\mu_k, \Sigma_k)$, where anatomically connected visible keypoints contribute weighted Gaussian estimations. This approach replaces binary visibility flags with continuous coordinate representations, enhancing inter-joint dependencies and structural coherence under partial observations.

- **Geometry-Aware Tokenizer**: A two-branch tokenization architecture that jointly learns discrete pose tokens and continuous position encoding maps, resolving spatial ambiguities in compositional part representation through explicit geometric priors.

- **SE-Driven Multimodal Fusion**: A parameter-efficient feature interaction module that unifies SE-based channel attention with coordinate-image feature fusion, enhancing occlusion reasoning by dynamically modeling structural-visual dependencies with minimal computational overhead.

- **Hierarchical Anatomical Regularization**: A body-part stratified loss function incorporating anatomical grouping constraints (head-torso, upper limbs, lower limbs) with differentiated weighting coefficients, enforcing biomechanically consistent pose estimation through joint coordinate optimization.

Our framework preserves PCT's efficiency in handling occlusions while significantly improving structural coherence and localization precision. Experiments demonstrate state-of-the-art performance on occluded scenarios.

## 2 RELATED WORK

### 2.1 HUMAN POSE ESTIMATION

2D human pose estimation research primarily bifurcates into multi-person and single-person estimation methodologies. Current multi-person approaches generally fall into three categories: top-down, bottom-up, and single-stage detection methods.

In top-down methodologies such as DeepPose, CPN, and AlphaPose Toshev & Szegedy (2014); Chen et al. (2018); Fang et al. (2017), the pipeline initially detects all individuals in the image through bounding box localization before conducting isolated pose estimation for each cropped instance. This sequential strategy ensures dedicated processing per subject but introduces computational redundancy during person detection. Bottom-up counterparts represented by algorithms like OpenPose operate inversely: they first detect all body keypoints indiscriminately, then cluster them into distinct person instances using association mechanisms such as Part Affinity Fields. Single-stage approaches exemplified by the Hourglass model Newell et al. (2016) integrate detection and estimation into end-to-end frameworks, eliminating intermediate processing steps. For single-person pose estimation, methodologies diverge into regression-based and heatmap-based paradigms.

**Regression Approaches:** Early works directly regress joint coordinates from images through fully-connected networks. Regression techniques directly predict keypoint coordinates through nonlinear mappings. While computationally efficient with implementations like DeepPose utilizing cascaded regression layers, DeepCut Pishchulin et al. (2016) addressing occlusion through integer programming, PIL Nie et al. (2018) enforcing anatomical limb constraints, and MSPN Li et al. (2019) refining through multi-stage processing, these methods exhibit inherent limitations. Despite advancements, regression-based systems plateau in accuracy due to coordinate regression's ill-posed nature and inability to model spatial uncertainty.

**Heatmap-based Approaches:** Heatmap-based methods Zhang et al. (2020); Nibali et al. (2018) currently dominate the field due to superior spatial precision, generating per-joint probability distributions where pixel intensity corresponds to location confidence. This approach preserves spatial relationships and effectively leverages convolutional architectures. Notable implementations include the Hourglass network with its symmetric encoder-decoder design, Simple Baseline Xiao et al. (2018) employing efficient deconvolutional upsampling, and HRNet Sun et al. (2019) maintaining high-resolution feature maps throughout processing. Nevertheless, significant challenges persist; while achieving state-of-the-art results, heatmap methods remain susceptible to spatial prediction failures under severe occlusions and require computationally expensive post-processing such as non-differentiable coordinate decoding via argmax operations. These fundamental gaps in handling occlusion and computational inefficiency motivate our cross-modal completion approach for robust pose estimation.

### 2.2 ATTENTION AND GATING MECHANISMS

Attention mechanisms Bahdanau et al. (2016); Chen et al. (2018) in deep learning emulate the information-filtering properties of the human visual system, empowering models to dynamically focus on critical input features and substantially enhance generalization capabilities. This approach proves particularly effective for processing long-sequence data—such as text, speech, and image sequences—by generating position-specific weights across spatiotemporal dimensions, thereby optimizing information extraction efficiency. The core operational paradigm relies on the Query-Key-Value triadic computation: similarity scores between Query and Key vectors generate attention weights, which then perform weighted aggregation of Value vectors to yield context-aware representations.

Beyond natural language processing Bojanowski et al. (2017); Raffel et al. (2020); Radford et al. (2021), attention mechanisms now serve as fundamental cross-modal fusion tools in multimodal tasks. In visual question answering, models like ViLBERT Lu et al. (2019) employ cross-modal attention to semantically align textual queries with relevant image regions; in video captioning, hierarchical attention mechanisms coordinate spatiotemporal features for precise motion trajectory modeling; in text-guided image synthesis, attention weight allocation enables semantically-controllable local rendering.

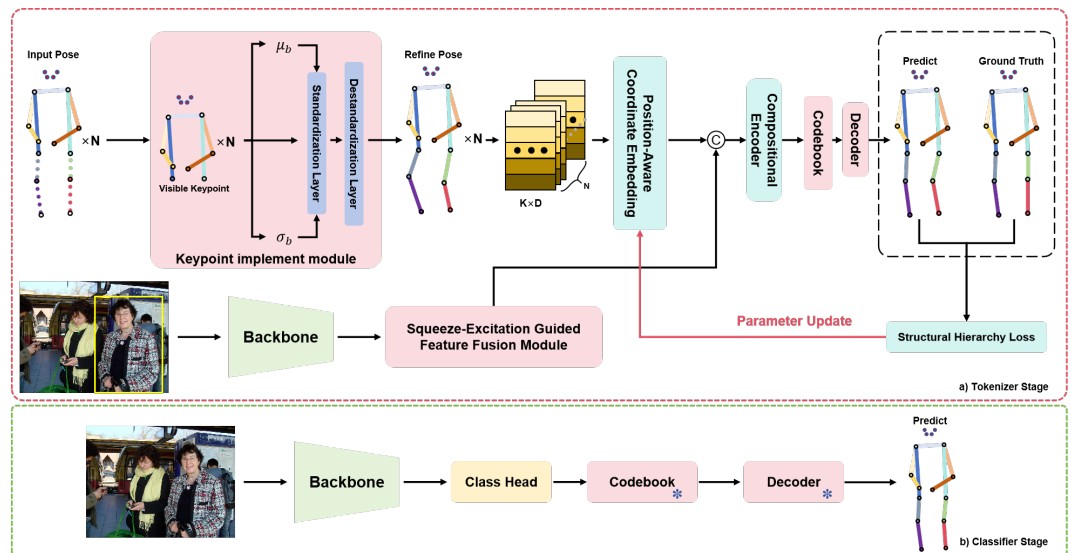

Figure 1: The overall training pipeline of our model consists of two stages (a, b). In stage (a), the tokenizer phase, the model jointly trains various module parameters using both keypoint information and image features. It leverages keypoint completion for prior information, incorporates spatial features, and ultimately represents the pose as a combination of discrete tokens. In stage (b), the classifier phase, the model freezes all module parameters trained in the previous stage and matches image features with corresponding tokens solely based on image information, finally decoding the final pose.

## 3 METHOD

In Section 3.1, we outline the architectural blueprint of our neural network alongside its hierarchical feature extraction modules. Subsequently, Section 3.2 introduces encoder enhancements through three synergistic innovations: a probabilistic keypoint completion module modeling occluded joints using Bayesian inference principles; a spatial feature integration strategy designed to mitigate ambiguity in spatial representations by adaptively fusing coordinate features with iteratively refined positional information; and a hierarchical attention mechanism adaptively reinforcing biomechanically correlated joint relationships through multi-scale feature recalibration. Section 3.3 proposes an anatomically-weighted loss function prioritizing biomechanically critical joints over peripheral ones through learned importance weighting.

### 3.1 PCT NETWORK ARCHITECTURE

**PCT (Pose as Compositional Tokens)** adopts a two-stage framework to model joint dependencies through structured token representations, enhancing robustness in occluded scenarios. Its core innovation lies in decomposing poses into compositional discrete tokens and mapping images to poses via a classification-based pipeline. The structure is detailed as follows:

#### 3.1.1 OVERVIEW OF TWO-STAGE FRAMEWORK

The PCT model comprises **representation learning** and **pose estimation stages**: Stage I (Representation Learning): This stage constructs structured pose representations via a compositional encoder $f_e(\cdot)$, which maps raw pose coordinates $\mathbf{G} \in \mathbb{R}^{B \times K \times D}$ ($K$ joints, $D$ dimensions) into $M = 16$ token features $\{\mathbf{t}_i\}_{i=1}^{M}$, each encoding interdependent joint substructures. These tokens are discretized via nearest-neighbor search in a shared codebook $\mathbf{C} \in \mathbb{R}^{1024 \times H}$ Vaswani et al. (2017), and a decoder $f_d(\cdot)$ reconstructs the pose $\hat{\mathbf{G}}$ from the quantized tokens $\{\mathbf{c}_{q(\mathbf{t}_i)}\}$. Two enhancements are integrated: 1) random joint masking to enforce structural completeness, and 2) fusion of joint-local image features with positional encodings to improve discriminability. The framework achieves sub-pixel reconstruction accuracy while compactly encoding pose variations. However, a limitation lies

in its lack of explicit spatial modeling and attention-based feature refinement, potentially hindering robustness to complex spatial occlusions or dynamic interactions. Moreover, as PCT directly infers missing keypoints without leveraging prior structural constraints or consistency checks, it may struggle to recover anatomically plausible poses when confronted with large-scale missing data or ambiguous configurations.

Stage II (Pose Estimation): Pose estimation is cast as a classification task. Image features from a backbone network are fed to a classification head to predict $M$ token categories. The decoder directly generates pose coordinates without post-processing.

This decoupled design models joint dependencies while simplifying the inference pipeline.

### 3.1.2 COMPOSITIONAL ENCODING AND QUANTIZATION

**Encoder Design:** Pose coordinates $G$ are first projected to higher dimensions via linear layers, then processed by MLP-Mixer Tolstikhin et al. (2021) blocks to fuse global joint dependencies, yielding $M$ token features $T = (t_1, t_2, \ldots, t_M) = f_e(G)$, where $t_i \in \mathbb{R}^H$. MLP-Mixer alternates channel-wise and spatial MLP operations to efficiently model interactions.

**Codebook Quantization:** A shared codebook $\mathcal{C} = [c_1, \ldots, c_V]^T \in \mathbb{R}^{V \times N}$ quantizes each $t_i$ to its nearest entry:

$$q(t_i) = \arg \min_j \|t_i - c_j\|_2 \tag{1}$$

Quantized tokens $\{c_{q(t_1)}, \ldots, c_{q(t_M)}\}$ are decoded by $f_d$ to reconstruct poses $\hat{G} = f_d(c_{q(t_1)}, \ldots, c_{q(t_M)})$. The decoder uses a shallow MLP-Mixer to invert the encoding process.

### 3.2 TECHNICAL MODIFICATIONS TO POSE ENCODING

**Gaussian-based Keypoint Completion:** Standard PCT models employ a masking operation $\mathbf{J}_{\text{masked}} = \mathbf{J} \odot \mathbf{V}$ to occlude invisible keypoints. While this mitigates spatial uncertainty interference, it results in inefficient utilization of critical information. Our method proposes a Gaussian distribution-based keypoint compensation mechanism, leveraging the distribution characteristics of keypoint coordinates from large-scale datasets to fit a multivariate Gaussian model $\mathcal{N}(\mu_{\text{pop}}, \Sigma_{\text{pop}})$. Through normalization design that eliminates scale/translation variations across individual poses (preventing domain shift in distribution parameter estimation), we infer global Gaussian parameters $\mu_b, \sigma_b$ from visible keypoint distributions, subsequently achieving conditional completion of invisible points and denormalization reconstruction. This outputs a complete pose hypothesis distribution $\hat{\mathcal{P}} \sim \mathcal{N}(\mu_{\text{cond}}, \Sigma_{\text{cond}})$.

The specific workflow proceeds as follows: First, visible keypoint screening:

$$\mathcal{J}_b^{\text{vis}} = \{\mathbf{J}_{b,k} \mid V_{b,k} = 1\} \tag{2}$$

The coordinate tensor $\mathbf{J} \in \mathbb{R}^{B \times K \times 2}$ and visibility mask $\mathbf{V} \in \{0,1\}^{B \times K}$ ensure parameters originate from actual observations. Statistical computation:

$$\mu_b = \frac{1}{|\mathcal{J}_b^{\text{vis}}|} \sum_{\mathbf{j} \in \mathcal{J}_b^{\text{vis}}} \mathbf{j} \tag{3}$$

$$\sigma_b = \sqrt{\frac{1}{|\mathcal{J}_b^{\text{vis}}|} \sum_{\mathbf{j} \in \mathcal{J}_b^{\text{vis}}} \|\mathbf{j} - \mu_b\|^2 + \epsilon} \tag{4}$$

where $\mathbf{j} \in \mathbb{R}^2$ denotes the 2D coordinate vector of a visible keypoint, $\epsilon = 10^{-6}$ prevents numerical singularity.

**To eliminate scale and translation differences across pose instances**, standardized transformation is performed:

$$\hat{\mathbf{J}}_{b,k} = \frac{\mathbf{J}_{b,k} - \mu_b}{\sigma_b} \tag{5}$$

establishing a geometrically invariant space. Invisible point estimation requires visible point cardinality computation $n_b^{\text{vis}} = \sum_{k=1}^{K} V_{b,k}$——this counting operation bears dual statistical significance: 1) Serves as distribution reliability indicator ($n_b^{\text{vis}} \geq 3$ enables minimal planar constraints); 2) Determines compensation strategy confidence level (high $n_b^{\text{vis}}$ employs precise local statistics, low $n_b^{\text{vis}}$ activates conservative global priors). Conditional completion follows:

$$\hat{\mathbf{J}}_{b,k} \leftarrow \begin{cases} \frac{1}{n_b^{\text{vis}}} \sum_{m:V_{b,m}=1} \hat{\mathbf{J}}_{b,m} & \text{if } V_{b,k} = 0 \ \wedge \ n_b^{\text{vis}} \geq 3 \\ \hat{\mathbf{J}}_{b,k} & \text{otherwise} \end{cases} \tag{6}$$

Under the distribution consistency hypothesis expressed by the conditional probability $p(J_{\text{occ}} \mid J_{\text{vis}}) \sim \mathcal{N}(\hat{\mu}_b, \hat{\Sigma}_b)$, the denormalization reconstruction is performed:

$$\mathbf{J}_{b,k}^{\text{est}} = \hat{\mathbf{J}}_{b,k} \cdot \sigma_b + \mu_b \tag{7}$$

outputs complete coordinate tensor $\mathbf{J}^{\text{est}} \in \mathbb{R}^{B \times K \times 2}$. This approach achieves probabilistic completeness through explicit modeling of pose distribution parameters, dynamically adapts to visible point counts (employing covariance-weighted precision modeling when $n_b^{\text{vis}} \geq 10$, isotropic Gaussian conservative compensation when $3 \leq n_b^{\text{vis}} < 10$), geometrically decouples translation($\mu_b$)/scale($\sigma_b$)/rotation attributes, and enhances occlusion robustness with $\mathcal{O}(BK)$ computational complexity.

**Position-Aware Coordinate Embedding:** Although raw coordinates capture geometric relationships, they lack explicit anatomical localization cues critical for distinguishing symmetric joints and similar local structures. We address this limitation by augmenting PCT with positional awareness through learnable anatomical priors.

We enhance joint coordinate embedding by incorporating anatomical priors through learnable positional encoding vectors $\mathbf{P} \in \mathbb{R}^{K \times d}$. The input features $\mathbf{G} \in \mathbb{R}^{B \times K \times 3}$ are constructed by concatenating the estimated keypoint coordinates $\mathbf{J}^{\text{est}} \in \mathbb{R}^{B \times K \times 2}$ (from the probabilistic completion module) with the visibility mask $\mathbf{V} \in \{0,1\}^{B \times K}$ along the last dimension. The modified joint representation for the $j$-th joint is formulated as follows:

$$\mathbf{E}_j = \text{Linear}(\mathbf{G}_j) + \mathbf{P}_j \tag{8}$$

$$\mathbf{P}_j = \mathbf{P}_j^{(\text{tri})} + \mathbf{P}_j^{(\text{learn})} \tag{9}$$

For each joint $j \in \{1, ..., K\}$, the embedding $\mathbf{E}_j \in \mathbb{R}^d$ is computed through two complementary components: 1) The coordinate vector $\mathbf{G}_j \in \mathbb{R}^D$ (Where $D = 3$ for 2D poses: each joint is represented as $(x, y, v)$, with $v \in [0,1]$ denoting visibility.) undergoes linear projection via Linear$(\cdot) : \mathbb{R}^D \to \mathbb{R}^d$ to capture geometric patterns; 2) A learnable positional encoding vector $\mathbf{P}_j \in \mathbb{R}^d$ is added to inject anatomical priors about the joint's typical spatial location (e.g., wrist vs. ankle).

The hybrid encoding $\mathbf{P}_j$ merges a fixed anatomical prior $\mathbf{P}_j^{(\text{tri})}$ with a trainable offset $\mathbf{P}_j^{(\text{learn})}$. Initialized from $\mathcal{N}(0, 0.02^2)$, $\mathbf{P}_j^{(\text{learn})}$ is updated via gradient descent to adaptively refine joint positioning. This dual-stream mechanism preserves biomechanical constraints while learning dataset-specific spatial variations.

**Squeeze-Excitation Guided Feature Fusion:** Standard PCT fusion equally weights feature channels, overlooking critical motion cues. We introduce channel-wise attention that dynamically recalibrates features, emphasizing discriminative patterns while suppressing redundancies.

We bring in chandunel-wise attention to the feature fusion process. After projecting joint embeddings through linear layers, we first refine the image feature maps $\mathbf{F} \in \mathbb{R}^{B \times C \times H \times W}$ using a Squeeze-and-Excitation (SE) block before concatenation:

$$\mathbf{F}' = \mathbf{F} \odot \text{sigmoid}\left(\mathbf{W}_2 \cdot \text{GELU}(\mathbf{W}_1 \cdot \text{GAP}(\mathbf{F}))\right) \tag{10}$$

where GAP($\cdot$) denotes the global average pooling, $\mathbf{W}_1 \in \mathbb{R}^{C/r \times C}$ and $\mathbf{W}_2 \in \mathbb{R}^{C \times C/r}$ form the attention bottleneck, with $r = 16$ as the reduction ratio. The refined features are then processed as:

We first reshape the feature maps $\mathbf{F}$ processed by Swin-Transformer to spatial dimensions $\mathbf{F}_{\text{sp}} \in \mathbb{R}^{B \times K \times H \times W}$, then applying squeeze-and-excitation attention, and finally flattening and projecting the attended features. This is mathematically represented as:

$$\mathbf{F}' = \text{SE}(\mathbf{F}_{\text{sp}}) \tag{11}$$

$$\mathbf{F}_{\text{proj}} = \mathbf{W}_e \cdot \text{vec}(\mathbf{F}') \tag{12}$$

The final stage concatenates this representation with embedding features:

$$\mathbf{F}_{\text{fused}} = [\mathbf{E}_{\text{embed}} \| \mathbf{F}_{\text{proj}}] \in \mathbb{R}^{B \times K \times (d+d')} \tag{13}$$

where $\mathbf{E}_{\text{embed}} \in \mathbb{R}^{B \times K \times d}$ is the **aggregated joint embedding tensor**, formed by stacking individual joint embeddings, while $d'$ denotes the projected joint feature dimension.

Our innovations in **Keypoint Completion**, **Position-Aware Coordinate Embedding**, and **Feature Fusion** advance pose representation through three principal contributions: 1) Keypoint Completion employs probabilistic modeling with Gaussian distributions to generate anatomically plausible predictions for occluded joints, enforcing biomechanical constraints that maintain skeletal coherence under severe occlusions; 2) Learnable anatomical priors explicitly resolve spatial ambiguities in symmetric joint localization, particularly enhancing left-right limb differentiation under partial observations; 3) A channel-wise attention mechanism dynamically prioritizes discriminative motion patterns while suppressing noise-corrupted features through global context modeling. These innovations collectively establish a more robust framework for handling complex spatial interactions and transient occlusions compared to conventional coordinate-based approaches.

### 3.3 Structural Hierarchy Loss

**Biomechanical Prior Guided Loss:** We propose a biomechanically informed loss function that adaptively weights joint prediction errors based on anatomical significance. The hybrid loss combines our original PCT reconstruction loss $\mathcal{L}_{\text{pct}}$ with a novel loss of biomechanical structure $\mathcal{L}_{\text{reg}}$ through learnable weight coefficients $(\alpha, \gamma)$:

$$\mathcal{L}_{\text{bpg}} = \alpha \mathcal{L}_{\text{pct}} + \gamma \mathcal{L}_{\text{reg}} \tag{14}$$

$$\mathcal{L}_{\text{pct}} = \text{smooth}_{L_1}(\hat{G}, G) + \beta \sum_{i=1}^{M} \| t_i - \text{sg}[c_{q(t_i)}] \|_2^2 \tag{15}$$

where $\mathcal{L}_{\text{reg}}$ imposes hierarchical anatomical constraints through region-specific error weighting, formulated as:

$$\mathcal{L}_{\text{reg}} = \sum_{j \in \mathcal{S}_{\text{ht}}} \omega_{\text{ht}} \| \hat{\mathbf{p}}_j - \mathbf{p}_j \|_2 + \sum_{j \in \mathcal{S}_{\text{arm}}} \omega_{\text{arm}} \| \hat{\mathbf{p}}_j - \mathbf{p}_j \|_2$$
$$+ \sum_{j \in \mathcal{S}_{\text{leg}}} \omega_{\text{leg}} \| \hat{\mathbf{p}}_j - \mathbf{p}_j \|_2 \tag{16}$$

where $\mathcal{S}_{\text{ht}} \subset \{1, ..., K\}$ denotes head-torso joint indices (e.g., head, neck, spine), $\mathcal{S}_{\text{arm}}$ and $\mathcal{S}_{\text{leg}}$ represent upper / lower extremity joints respectively, with anatomically calibrated weights $\omega_{\text{ht}}$, $\omega_{\text{arm}}$, $\omega_{\text{leg}}$ reflecting biomechanical significance levels.

## 4 Experiment

### 4.1 Experimental Setup

**Datasets.** We first evaluate our model on standard pose estimation benchmarks: the COCO dataset Lin et al. (2015). The COCO dataset contains over 160,000 images and 250,000 human instances,

Table 1: Performance comparison on COCO val set. The best results are shown in bold, and the second-best results are underlined. We conducted our experiments using a single NVIDIA RTX 3090 GPU with a batch size of 32. All our experiments were performed on the standard COCO dataset, and the reported accuracies represent the best performance achieved across comprehensive experiments.

| Method | Backbone | Input size | GFLOPs ↓ | Speed (fps) ↑ | COCO val2017 ↑ | | |
|---|---|---|---|---|---|---|---|
| | | | | | AP | AP$^{50}$ | AP$^{75}$ |
| SimBa. | ResNet-152 | 384 × 288 | 28.7 | 76.3 | 74.3 | 89.6 | 81.1 |
| PRTR | HRNet-W32 | 384 × 288 | 21.6 | 87.0 | 73.1 | 89.4 | 79.8 |
| TransPose | HRNet-W48 | 256 × 192 | 21.8 | 56.7 | 75.8 | 90.1 | 82.1 |
| TokenPose | HRNet-W48 | 256 × 192 | 22.1 | 52.9 | 75.8 | 90.3 | 82.5 |
| HRNet | HRNet-W48 | 384 × 288 | 35.5 | 75.5 | 76.3 | 90.8 | 82.9 |
| DARK | HRNet-W48 | 384 × 288 | 35.5 | 62.1 | 76.8 | 90.6 | 83.2 |
| SimCC | HRNet-W48 | 384 × 288 | 32.9 | 71.4 | 76.9 | 90.9 | 83.2 |
| HRFormer | HRFormer-B | 384 × 288 | 29.1 | 25.2 | 77.2 | 91.0 | 83.6 |
| ViTPose | ViT-Base | 256 × 192 | **17.9** | 113.5 | 75.8 | 90.7 | 83.2 |
| SimBa. | Swin-Base | 256 × 256 | 16.6 | 74.4 | 76.6 | **91.4** | 84.3 |
| Our approach | Swin-Base | 256 × 256 | 30.4 | **148.26** | **77.3** | 90.9 | 84.3 |

Table 2: Comparative analysis of model performance

(a) Performance comparison on OCHuman dataset

| Model | OCHuman (mAP) |
|---|---|
| HRFormer-s | 60.3 |
| PVTv2 | 58.5 |
| SWIN-t | 58.1 |
| ViTPose-s | 60.3 |
| ProbPose-s | 60.4 |
| ProbPose-s-DP | 61.4 |
| Our approach | **64.63** |

(b) Ablation study of main components (COCO val mAP)

| KeyC | CooE | SE | SH-L | mAP |
|---|---|---|---|---|
| | ✓ | ✓ | ✓ | 76.0 |
| ✓ | | ✓ | ✓ | 75.9 |
| ✓ | ✓ | | ✓ | 76.0 |
| ✓ | ✓ | ✓ | | 75.7 |
| ✓ | ✓ | ✓ | ✓ | 76.2 |

each annotated with 17 keypoints. Its training set includes approximately 118,000 instances, validation set 5,000 instances, and test set 40,000 instances.

Subsequently, we test on occlusion-specific benchmarks: OCHuman.This multi-dataset strategy enables dual evaluation: quantifying baseline pose estimation capability on standard benchmarks, while specifically assessing performance improvements in handling occlusions through our keypoint completion and feature fusion modules on occlusion datasets.

**Evaluation metrics.** For the COCO dataset, we employ standard keypoint detection metrics: Average Precision (AP) and Average Recall (AR) calculated at Object Keypoint Similarity (OKS) thresholds. Specifically, we report AP (averaged over 10 OKS thresholds), AP50 (OKS = 0.50), AP75 (OKS = 0.75), and APM / L (medium / large scale objects).

### 4.2 IMPLEMENTATION DETAILS

We adopt a top-down detection pipeline throughout the training process. During training, GT bounding boxes are utilized to precisely crop human instances (padding factor=1.25), ensuring accurate keypoint localization learning. For inference testing, we replace GT boxes with detection boxes generated by a pre-trained Faster R-CNN model He et al. (2015) to simulate real-world scenarios.

For image feature extraction, we implement Swin-Transformer V2 Liu et al. (2021) backbone pre-trained on ImageNet-22K, processing input images at 256×256 resolution (heatmap size 64×64), with frozen first 5 stages.

## 4.3 RESULTS ON COCO

Table 1 shows the evaluation results of various top-down methods on COCO val2017 set. Our model achieves superior or comparable performance in terms of detection accuracy compared to other methods. Specifically, our model achieves the highest AP of 77.3, outperforming the second-best HRFormer (77.2). For $AP^{75}$, our model achieves 84.3, sharing the highest score with SimBa Patro & Agneeswaran (2024). In terms of inference efficiency, our model demonstrates exceptional speed performance with 148.26 fps, which is significantly faster than the second-best ViTPose Dosovitskiy et al. (2021) at 113.5 fps. While our GFLOPs (30.4) is moderate, the substantial speed advantage makes our model particularly attractive for real-world applications. For reference, most HRNet-based models operate at speeds below 80 fps, highlighting our model's significant efficiency improvement.

## 4.4 RESULTS ON OCHUMAN

To evaluate our method's performance in occluded scenarios, we conducted experiments on the OCHuman dataset. In Table 2 (a), we denote the best performance in bold and the second-best results with underlines. The results demonstrate that our approach achieves substantial improvements over other methods: compared to the method ProbPose-s-DP Purkrabek & Matas (2024), our approach reaches 64.63% mAP, representing a significant improvement of 3.23 percentage points. Notably, when compared to methods with commonly used backbones such as HRFormer-s Xiao et al. (2018) and ViTPose-s (60.3% mAP), our approach exhibits superior robustness in heavily occluded scenarios, with performance gains exceeding 4 percentage points. This remarkable improvement validates the effectiveness of our approach in handling complex occlusion scenarios.

## 4.5 ABLATION RESULTS

To thoroughly evaluate the effectiveness of our proposed framework, we conducted comprehensive ablation studies on several key components: Keypoint Completion (KeyC), Coordinate Embedding (CooE), Squeeze-Excitation Guided Feature Fusion (SE), and Structural Hierarchy Loss (SH-L). All experiments were performed on the COCO validation set using ground truth bounding boxes to ensure fair comparison.

In our systematic investigation, we first examined the impact of Keypoint Completion by removing it from the framework. This modification significantly impacted the model's prediction guidance capability. As illustrated in Table 3, the model's performance decreased to an mAP of 76.0%, demonstrating a notable drop compared to our complete architecture. This decline can be attributed to the absence of crucial prior information, which consequently required extended training time for the model to achieve convergence.

Subsequently, we investigated the role of Coordinate Embedding by removing this component from our architecture. The experimental results revealed that without spatial information encoding, the model's accuracy dropped to 75.9% and exhibited substantially slower convergence during the training process. In our third ablation experiment, removal of the SE module eliminated the channel-wise attention mechanism. While the immediate impact on accuracy metrics was modest, this modification significantly disrupted the inter-modal feature relationships. Finally, we examined the impact of our proposed loss design, with the model achieving 75.7% mAP when reverting to traditional loss functions.

## 5 CONCLUSION

In this work, we present CM-PCT, a novel structure-aware framework for human pose estimation that integrates spatial relationships between keypoints with probabilistic modeling of anatomical joint configurations. Our approach features a specialized loss function designed to leverage the inherent hierarchical structure of human anatomy, achieving performance comparable to state-of-the-art methods while providing better interpretability.

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
