# OpenReview forum: "Compositional Token Modeling for Occlusion-Robust Human Pose Estimation​"
_ICLR.cc/2026/Conference — ICLR 2026 Conference Withdrawn Submission_

### Official Review · Reviewer_gJ2U · 2025-10-31

**Soundness:** 2
**Presentation:** 2
**Contribution:** 2
**Rating:** 2
**Confidence:** 4

**Summary:**

This paper aims to address the robustness issue of pose estimation under occlusion scenarios. Based on the previous VAE-based PCT model, the authors propose an improved framework named CM-PCT. The main improvements include:

1) Joint completion in the pre-training stage based on Gaussian distribution,

2) The introduction of both image and joint features in the encoder for feature fusion, and

3) Several technical refinements such as coordinate embedding and weighted loss.

The paper presents its ideas clearly, and the design of the pre-training stage is somewhat innovative. However, the major concern is the lack of experimental evidence, both qualitative and quantitative, to substantiate the claimed robustness improvement under occlusion conditions. As a result, the contribution remains mostly descriptive without sufficient data support.

Moreover, since the proposed method is built upon the PCT model, a direct performance comparison with PCT is expected. Without experimental results, it is unclear what these specific impacts of the proposed modifications actually bring to the baseline PCT model.

Finally, the writing and presentation could be improved. For example, in the tables, all compared methods should include proper citations to help readers and reviewers locate the referenced works more easily.

**Strengths:**

The design of the pose completion during pre-training and the integration of image features are well-motivated.

**Weaknesses:**

The paper is not sufficiently complete or detailed, and most of its contributions remain descriptive rather than being supported by experimental evidence.

**Questions:**

Refer to the Summary part.

---

### Official Review · Reviewer_g5BC · 2025-10-31

**Soundness:** 1
**Presentation:** 2
**Contribution:** 1
**Rating:** 2
**Confidence:** 4

**Summary:**

This paper proposes CM-PCT for occlusion-robust human pose estimation. The method builds on the Pose as Compositional Tokens (PCT) framework by introducing Gaussian-based keypoint completion, SE-based cross-modal fusion, and anatomically guided loss. Experiments on COCO and OCHuman datasets show improvements over several baselines.

**Strengths:**

1. This paper is easy to follow.

**Weaknesses:**

1. The core idea of this paper is almost identical to the PCT[1] framework. The main technical additions (e.g., Gaussian keypoint completion, position-aware embedding, and SE fusion) appear to be incremental modifications rather than a fundamentally new perspective. More importantly, The paper does not provide any direct comparison with the original PCT baseline. A side-by-side quantitative and qualitative analysis with PCT is essential for evaluating novelty and merit. The author should report the model performance on COCO test-dev, MPII, H36M, and test PCT on OCHuman.
2. As shown in the PCT paper, the performance on COCO val2017 is higher than this paper. So the effectiveness of technical additions is questionable.
3. The paper reports quantitative metrics but completely lacks visualizations or qualitative analysis. Given the claim of “occlusion robustness,” visual comparisons of predicted poses under partial occlusion, as well as attention or token activation maps, are necessary to substantiate the claim. Without such evidence, the method’s claimed interpretability and structural reasoning advantages remain speculative.

[1] Human Pose as Compositional Tokens.

**Questions:**

See weaknesses.

---

### Official Review · Reviewer_8p6Y · 2025-11-01

**Soundness:** 2
**Presentation:** 3
**Contribution:** 2
**Rating:** 4
**Confidence:** 4

**Summary:**

This paper proposes CM-PCT, an enhanced framework for 2D human pose estimation designed specifically to handle severe occlusions. The method builds upon the existing two-stage PCT (Pose as Compositional Tokens) architecture, which first learns a discrete codebook of "pose tokens" representing anatomical substructures and then trains an image-based model to classify these tokens.
The authors propose four innovations towards this. First, Gaussian-based keypoint completion to infer occluded joints probabilistically. Second, position-aware coordinate embeddings to inject anatomical priors and spatial structure. Third, squeeze-and-excitation attention for adaptive cross-modal fusion of image and coordinate features. Fourth, hierarchical anatomical regularization loss to enforce biomechanical consistency through body-part-specific supervision.

Experiments on COCO and OCHuman datasets demonstrate competitive or superior accuracy under occlusion, while maintaining efficiency (148 fps on a single GPU).

**Strengths:**

1. The paper is well-motivated, identifying an underexplored issue in occlusion robustness by separating structural reasoning from feature corruption recovery.
2. The Keypoint Completion module seems to be the most significant contribution. The paper introduces a mathematically principled approach to infer occluded joints via conditional Gaussian modeling. Instead of forcing the model to learn from corrupted data (poses with zeroed out joints), the authors first repair the data using a probabilistic model. This approach of providing a complete, plausible pose to the tokenizer seems to be highly effective.
3. The integration of learnable positional embeddings and triangulated anatomical priors (Eq. 8, 9) helps differentiate symmetric or visually similar parts (example left/right limbs).
4. The SE inspired fusion block adaptively reweights RGB features and coordinate embeddings, providing fine grained feature interaction between modalities. Also, the design is parameter-efficient and maintains high inference speed.
5. The claims of occlusion robustness are supported by the results. While the performance on the OCHuman dataset is superior by 3.23 mAP, the performance on COCO is competitive with prior arts and running at 148 fps demonstrates good optimization for real-time deployment.

**Weaknesses:**

1. Although the extensions are meaningful, much of the architecture still heavily depends on PCT’s compositional framework. The Gaussian completion and geometry-aware tokens can be viewed as incremental refinements rather than innovations.
2. The Gaussian-based imputation seems hand-crafted (conditional on visible joints with fixed priors). It lacks rigorous justification or comparison with learned probabilistic models (e.g., VAEs).
3. A major weakness of this paper is the ablation study in Table 2 (b). The quantitative impact of each of the four key innovations is shown to be minimal on the COCO dataset as shown by the tiny drops. This contradicts the author’s claim that these components are critical.
4. The authors perform ablations on COCO, a general-purpose benchmark, but not on OCHuman. The paper's main claim is occlusion robustness, and the OCHuman dataset is where the method truly shines.

**Questions:**

1. Are the positional priors P_tri manually designed or derived from data statistics?
2. How does the fixed + learnable embedding compare to standard learned positional embeddings such as sinusoidal or learned 2D embeddings?
3. Could this hierarchical loss cause bias toward certain joints such as upper body dominance?
4. Why is the final number in Table 2 (b) not the same as the performance reported for the model in Table 1?
5. Please answer questions arising from the Weakness section as well.

---

### Official Review · Reviewer_g9G3 · 2025-11-01

**Soundness:** 1
**Presentation:** 2
**Contribution:** 1
**Rating:** 0
**Confidence:** 4

**Summary:**

This paper introduces CM-PCT, an extension of the Pose as Compositional Tokens (PCT) [1] framework, designed to enhance robustness against occlusion. The proposed approach integrates four key components: (1) gaussian-based keypoint completion (§3.2), (2) position-aware embedding (§3.2), (3) SE-guided multimodal feature fusion (§3.2), and (4) structural hierarchical loss for anatomically informed training (§3.3). Experimental results on COCO and OCHuman benchmarks demonstrate only modest improvements over existing methods (e.g., +0.1 AP on COCO (Table 1) and +3.2 mAP on OCHuman (Table 2 (a))).

[1] Z. Geng et al., Human Pose as Compositional Tokens, in CVPR, 2023.

**Strengths:**

+ Handling occlusion is an important and challenging problem, and the proposed approach could be viewed as a reasonable engineering attempt to mitigate this issue.

**Weaknesses:**

- The overall framework is largely derived from PCT [1], and all the proposed components are well-known and widely used techniques. Consequently, the novelty of this work is quite limited.
- As shown in Table 1, the improvement on the COCO dataset is marginal. Although the results on OCHuman (Table 2(a)) demonstrate some advantage over prior methods, the choice of comparison baselines appears somewhat arbitrary, and several state-of-the-art methods are missing from the comparison (at the very least, PCT itself should be included).
- The literature review in §2 relies heavily on outdated references. More recent approaches should be discussed to better position the contribution within the current research landscape.
- As far as I understand, the "two-branch tokenization architecture" (Line 91) merely introduces learnable positional embeddings (§3.2). From this perspective, the term "Geometry-Aware Tokenizer" (Line 91) appears somewhat overstated.

**Questions:**

Please refer to the Weakness section.

---

### Note · Authors · 2025-11-13

I have read and agree with the venue's withdrawal policy on behalf of myself and my co-authors.